# Comparative Study of HVOF Cr$_3$C$_2$–NiCr Coating with Different Bonding Layer on the Interactive Behavior of Fatigue and Corrosion

**Bing He** [1,2,3], **Lijie Zhang** [1,*], **Xiao Yun** [2,3], **Jing Wang** [2,3], **Guangzhi Zhou** [2,3], **Zhikai Chen** [2,3] and **Xiaoming Yuan** [1,*]

1   Hebei Provincial Key Laboratory of Heavy Machinery Fluid Power Transmission and Control, Yanshan University, Qinhuangdao 066000, China; hebing5280@163.com
2   Jiangsu Xuzhou Construction Machinery Research Institute Co., Ltd., Xuzhou Construction Machinery Group, Xuzhou 221004, China; yunxiao0726@163.com (X.Y.); jingwang0822@126.com (J.W.); 13852092502@163.com (G.Z.); chenzhikai2004@126.com (Z.C.)
3   State Key Laboratory of Intelligent Manufacturing of Advanced Construction Machinery, Xuzhou 221004, China
*   Correspondence: zhangljys@126.com (L.Z.); xiaomingbingbing@163.com (X.Y.)

**Abstract:** In order to improve material service life under a fatigue and corrosion coupling environment, a high-velocity oxygen fuel (HVOF) Cr$_3$C$_2$–NiCr coating with a bonding layer was prepared. The objective was to obtain the optimum bonding layer for the HVOF Cr$_3$C$_2$–NiCr coating, which included a laser cladding (LC) Ni625 layer, extreme high-speed laser material deposition (EHLA) Ni625 layer and HVOF NiCr layer. Fatigue properties of the samples with various bonding layers were investigated by means of a four-point bending fatigue test. Electrochemical impedance spectroscopy (EIS) and the salt spray test were executed after the bending fatigue test to simulate the interactive effect of fatigue and corrosion atmosphere. Failure surfaces were characterized by scanning electron microscopy (SEM) and an energy-dispersive spectrometer (EDS) to indicate the details of corrosion products. Corrosive behaviors of samples were adequately demonstrated according to the results, which included the curves of potentiostatic polarization, impedance magnitude and phase degree, and corrosion products. The result showed that the cycles of perforative cracking for the sample with the EHLA Ni625 bonding layer was almost three times than that of the sample with the HVOF NiCr layer. The magnitude of EIS reduced from ~10$^5$ to ~10$^3$ for the sample after BFT. Eventually, the main improvement mechanism of the HVOF Cr$_3$C$_2$–NiCr coating with the EHLA Ni625 bonding layer was attributed to the grain refinement of the bonding layer and performed a good level of metallurgical bonding with the substrate.

**Keywords:** high-velocity oxygen fuel; laser cladding; extreme high-speed laser material deposition; bonding layer; bending fatigue; corrosion



## 1. Introduction

Hydro cylinders were the key components of a hydraulic system, which can effectively and steadily convert hydraulic energy to mechanical energy. As a result, hydro cylinders were widely used in marine equipment, construction machineries, mining machineries, etc. Since the frequent reciprocating motion or the prolonged exposure to air, the surface performance of piston rod directly affected the service life of the whole hydraulic system. High-velocity oxygen fuel was widely used to reinforce the surface properties because of the advantage of high flame velocity, high spray particle velocity and high kinetic energy [1]. Hence, the piston rod with the Cr$_3$C$_2$–NiCr coating fabricated by HVOF presented excellent performance in the coal mine, desert, ocean and oil field atmosphere [2–6]. T. Varis [7] et al. compared the fatigue performance of Cr$_3$C$_2$–NiCr coatings by different kerosene and

oxygen levels. Increased fatigue resistance of the coated material was attributed to the substantial compressive residual stresses that hindered it. However, amounts of unexpected porosity and unmelted particles were discovered at the interior of HVOF coatings, which were usually considered to be the potential failure source. Hence, it can be obviously proven that the HVOF coating is not a suitable under-fatigue condition, although it can provide sufficient wear and corrosion resistance. R.C. Souza [8] and Mitra Akhtari Zavareh [9] et al. compared the influence of $Cr_3C_2$–25NiCr and WC–10Ni coatings applied by the HVOF process and hard chromium electroplating on the fatigue strength, abrasive wear and corrosion resistance. It indicated that the HVOF $Cr_3C_2$–25NiCr coating had higher salt spray resistance in comparison with the chromium electroplating. Erwin Mayrhofer [10] et al. studied the cracking behavior of $Cr_3C_2$–25 (Ni20Cr) and WC–20$Cr_3C_2$–7Ni coatings carried out 3-point bending tests by Acoustic Emission (AE) monitoring. The AE monitoring result revealed a superior resistance against cracking in the WC–$Cr_3C_2$–Ni coatings compared with $Cr_3C_2$–NiCr.

In previous decades, composite coating had been widely investigated owing to its excellent combination properties. Henao [11] pointed out that in vitro bioactivity and electrochemical interactions of HVOF-Sprayed HAp/$TiO_2$ graded coatings had biocompatibility, adhesion strength and corrosion resistance. Bobzin [12] studied the thermal shock behavior of different multilayer thermal barrier coatings (TBCs) made from $La_2Zr_2O_7$ (LZ), $ZrO_2$-7 wt% $Y_2O_3$ (YSZ) and a freestanding LZ-YSZ coating by plasma spraying. The result shows that the compressive stresses in the LZ top layer decreased with an increasing coating thickness of the YSZ interlayer. Watanabe [13] studied the fracture behavior of WC–Co/copper multilayer coatings and demonstrated that the coating with higher volume fraction of copper (62%) exhibited more than two times higher work of fracture and better bending strength than the monolithic WC–Co coatings. Under fatigue condition, Erwin Mayrhofer et al. [10] stated that the moment the crack tip plastic zone reaches the interface between intermediate layer and substrate, the crack will be arrested, deflected and delamination may occur. Evidently, composite coating performed better than that of single-surface coatings, which could introduce a way to further improve coating properties.

Nevertheless, according to the above-mentioned method (fabricating the bonding layer by HVOF or plasma spraying), an enormous difference of element compositions between coatings and substrate was evidenced which leads to poor adhesion. Thus, the fatigue crack preferred to initiate at the interface of coating and substrate under fatigue condition [14] as a result of the maximum shear stress arising at the interface [15]. Therefore, the existence of bonding layer was considered to be prone to result in fatigue failure. Essentially, the improvement mechanism on the fatigue property for HVOF $Cr_3C_2$–NiCr coating with the bonding layer had not been carried out and discussed in previous studies.

Furthermore, the corrosion property of material was suggested to be significantly improved as result of grain refinement. Mohammad Islam et al. [16] confirmed that the addition of $SiO_2$ nanoparticles modifies deposit morphology through grain refinement, a reduction in the surface roughness and minimization of surface porosity in the nanocomposite coatings. Thus, the coating showed superior corrosion resistance offered by orders of magnitude upon $SiO_2$ incorporation. Additionally, Masoud Moshtaghi [17] indicated that the high grain boundary density in the nanostructured coating led to the presence of a larger number of strong H traps resulting in the lower diffusible H content. By the refinement of grain, the grain boundary density was subsequently increase, which improved the corrosion resistance of coating as response.

Since the 1960s, laser technology had been widely used in various fields, in particular for material manufacturing, due to its numerous advantages: high power density, low dilution rate and narrow heat-affected zone. Moreover, the surface properties were modified without changing the bulk material properties [18,19]. In addition, the processed region had a metallurgical bond with substrate [20,21], and the density of laser cladding layer was almost similar to bulk material, which gave excellent fatigue resistance. Hence, laser cladding layer was considered to be an optimal bonding layer, which could not only delay

the occurrence of fatigue defects, but also prevented corrosive elements from reaching the matrix surface. Luo [22] studied the bonding layer between the ceramic layer and substrate on thermal barrier coatings (TBCs) were MCrAlY coat by laser cladding processing. Laser cladding acting as metallurgical bonding had two disadvantages. However, the coating surface had a large roughness after laser cladding and the turning process was necessary before HVOF. What is more, uneven distribution of the residual stress and the big size of heat affected zone may be cause the deformation of a workpiece. In recent years, with the rapid development of new surface technology, especially extreme high-speed laser material deposition (EHLA) had been widely used in surface engineering [23,24]. However, the bonding layer fabricated by EHLA for HVOF $Cr_3C_2$–NiCr surface coating was rarely researched.

In previous studies, $Cr_3C_2$–NiCr coating fabricated by HVOF showed strong corrosive wear resistance [25–27] and laser cladding Ni625 coating exhibited excellent corrosive wear resistance [28]. In this study, the author intended to combine the advantages of both layers to improve service life under the combination of fatigue and corrosion conditions. To compare the effect of different bonding layers, surface failure behavior of HVOF $Cr_3C_2$–NiCr coating with three types of bonding layers, which are the HVOF NiCr bonding layer, laser cladding (LC) Ni625 bonding layer and extreme high-speed laser material deposition (EHLA) Ni625 bonding layer, were experimentally investigated. Microstructures, EIS and surface failure profiles of fabricated coatings were observed and compared, respectively. Subsequently, the mechanism of service life extension under the combination of fatigue and corrosion conditions was discussed.

## 2. Experimental Procedures

### 2.1. Experimental Materials

The base material was hardened and tempered 42CrMo steel and its chemical composition was illustrated in Table 1. With standard methods, the yield strength of the base material was 750 MPa. The $Cr_3C_2$–NiCr powder used for HVOF coating was a mixture of 25% NiCr and 75% $Cr_3C_2$ in weight (Figure 1a). The NiCr powder (Figure 1b) was composed of 80% Ni and 20% Cr in weight. It should be noted that the powder used for HVOF technology was ranged from 15 μm to 45 μm in demension. The laser cladding powder was Ni625, and its chemical composition was illustrated in Table 2. Dimensions of the powder used for LC (Figure 1c) and EHLA (Figure 1d) ranged from 53 μm to 108 μm and 20–45 μm, respectively.

**Table 1.** Chemical properties of 42CrMo steel (wt%).

| C | Cr | Mo | Si | Mn | Fe |
|---|---|---|---|---|---|
| 0.38–0.43 | 0.8–1.1 | 0.15–0.25 | 0.15–0.35 | 0.75–1.0 | Bal. |

**Table 2.** Chemical properties of Ni625 powder (wt%).

| Fe | C | Cr | Mo | Si | Mn | Cu | Ti | Nb | Ni |
|---|---|---|---|---|---|---|---|---|---|
| 5 | 0.1 | 21.5 | 9 | 0.5 | 0.5 | 0.5 | 0.4 | 4 | Bal. |

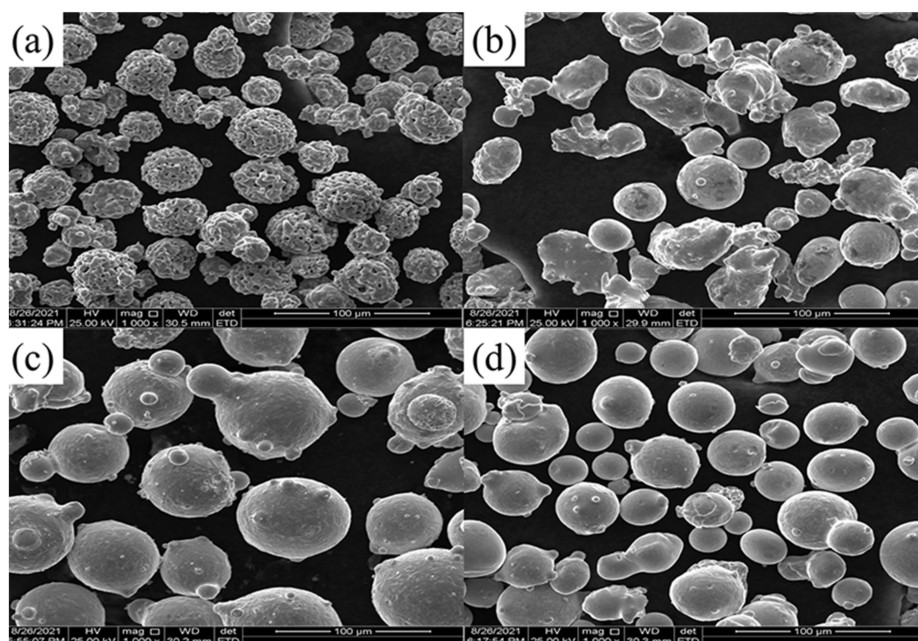

**Figure 1.** The SEM of used metal powder: (**a**) $Cr_3C_2$–NiCr; (**b**) NiCr; (**c**) Ni625 of LC and (**d**) Ni625 of EHLA.

### 2.2. Experimental Methods and Samples Preparation

The substrate was a 120 mm × 15 mm × 5 mm in dimension. To study the effect of the bonding layer on the interaction of fatigue and corrosion, three types of composite coatings were prepared, and the manufacturing processes are shown in Table 3. In the present paper, the grit-blasting was carried out by an automatic compression sandblasting machine with dimensions of 24 mesh white corundum. Both the NiCr bonding layer and the $Cr_3C_2$–NiCr surface layer were fabricated by the HVOF system (JET KOTEIII, Stellite, Latrobe, PA, USA). A laser cladding system (TruDisk 4002, Trumpf, Ditzingen, Germany) with a laser spot of 5 mm in diameter was utilized to prepare the LC Ni625 bonding layer. Additionally, the EHLA Ni625 bonding layer was fabricated by an EHLA system (LDF 4000, Laserline, Mullheim, Germany) with a laser spot of 1 mm in diameter. The various parameters of layer treatment processes were listed in Table 4. The layers were prepared by optimal process parameters on the basis of previous work. It should be noticed that there was no grit-blasting treatment following the EHLA process, owing to the EHLA layer presenting excellent surface roughness.

After surface treatment, the prepared coatings underwent grinding until the roughness reaching Ra 0.4 μm to reduce machining tool marks. For microstructure analysis, standard methods of metallography were carried out. The aqua regia (3:1, $v/v$, HCl to $HNO_3$) was used as a corrosive liquid due to high corrosion resistance of the prepared layers. Microstructures of the bonding layer and the surface coating were characterized by an inverted metallographic microscope (DMI5000M, Leica, München, Germany). According to metallographic method, the porosity of HVOF coating was analyzed. Surface hardness was measured by a Vickers hardness tester (model 5104, Buehler Co., Ltd., Sacramento, CA, USA) with a load of 300 g and a dwell time of 15 s. To ensure the accurate of the microhardness, three repetitive tests were executed for each location. Moreover, eight indentations, spaced 0.15 mm apart, were performed at cross-sections of the coatings along the depth direction.

**Table 3.** Process routing of experiment sample.

| Sample | Manufacturing Process |
|--------|----------------------|
| 1 | Grit-blasting→HVOF NiCr→HVOF $Cr_3C_2$–NiCr |
| 2 | LC→turning→grit-blasting→HVOF $Cr_3C_2$–NiCr |
| 3 | EHLA→grit-blasting→HVOF $Cr_3C_2$–NiCr |

**Table 4.** Technology parameters of grit-blasting, HVOF, LC and EHLA.

| Process | Parameters | Values |
|---------|-----------|--------|
| Grit-blasting | Air pressure (MPa) | 0.7–0.9 |
| | Blalting distance (mm) | 200 |
| | Blalting speed (mm/s) | 200 |
| | Blasting angle (°) | 90 |
| HVOF | Oxygen flow (SCFH) | 950–975 |
| | Propylene flow (SCFH) | 115–130 |
| | Powder feed rate (g/min) | 60–65 |
| | Spray distance (mm) | 180 |
| | Spray speed (mm·min$^{-1}$) | 1000–1500 |
| | Layer thickness (mm) | 0.1 |
| LC | laser power, W | 3800 |
| | scanning speed, mm/s | 40 |
| | overlap ratio | 50% |
| EHLA | laser power, W | 2200 |
| | scanning speed, mm/s | 300 |
| | overlap ratio | 50% |

*2.3. Dynamic Bending Tests*

In order to study the fatigue behavior of the sample with various coatings, the bending fatigue tests (BFT) were executed by a fatigue test system (MTS Flex Test60, MTS, Eden Prairie, MN, USA), dynamic servo control system (Dyna 900 DTB-S, Dailong, Hangzhou, China) and self-made four-point fatigue bending test unit (as shown in Figure 2a). The distance between the support points was 90 mm and the stress ratio was 0.1. According to the Det Norske Veritas (DNV) test standard, the test loading was determined to be 90% of theoretical yield strength of the substrate and the frequency was ranged of 0.25–1 Hz. As shown in Figure 2b, a strain gauge was glued on the surface. A dynamic signal test and analysis system (DH5922D, Donghua tester, Taizhou, China) was used to measure bending stress of the tested coating in real time. For accuracy of the experiment, three repetitive tests were performed for each group of samples. During the experiment, the testing process was interrupted to detect situation of surface crack every 500 cycles until to 2000 cycles. Once the visible fatigue crack was occurred, the experiment was terminated.

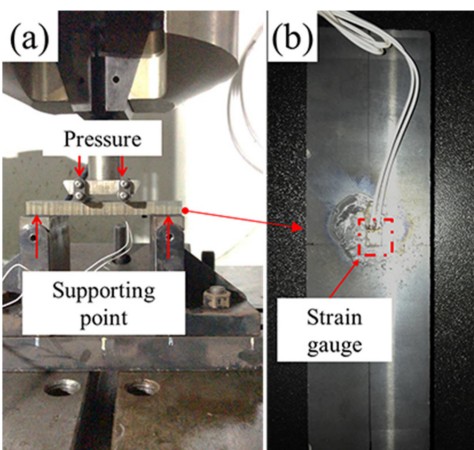

**Figure 2.** (**a**) The equipment of dynamic bending fatigue and (**b**) the details of the tested sample.

*2.4. Electrochemical Measurements and Salt Spray Corrosion Tests*

The electrochemical behavior of the coatings was investigated by potentiostat polarization curves and EIS. The potentiostat polarization test was carried out in line accordance with the requirement of Det Norske Veritas (DNV) test standard (DNV 2009-3296, 2009) in present paper. The objective of the test was to establish the degree of porosity and microcracks penetrating the coating down to the substrate. Open circuit potential was monitoring during 72 h until the system reached equilibrium. The test duration of samples No. 1, No. 2 and No. 3 was 500 h at $-350$ mV.

To identify the influence of bending fatigue on the corrosion behavior, the electrochemical impedance spectroscopy (EIS) was carried out before and after the four-point bending test. The electrolyte solution was artificial seawater. The electrochemical cell was placed in a Faraday cage to avoid interference from external electromagnetic fields and stray currents. The tests were measured to monitor the fatigue-corrosion process by an electrochemical station under 25 °C (ASTM D1141, Chenhua, Shanghai, China). The frequency range of EIS was from $10^{-2}$ Hz to $10^5$ Hz.

To study the corrosion failure behavior of samples with or without fatigue test, a contrastive experiment was designed. Two sets of same samples including No. 1–No. 3 were prepared. Salt spray corrosion test following bending test was carried out on one set of samples, whereas merely the salt spray corrosion test was carried out on the other set of samples. A neutral salt spray (NSS) corrosion test was carried out according to the standard method of ISO 9227-2006. After the corrosion test, the corrosive surface was investigated by means of tungsten filament scanning electron microscopy (SEM, Inspect S50, FEI, Hillsboro, OR, USA).

**3. Results and Discussion**

*3.1. Details of Deposition Layers*

The cross-sections of NO. 1–3 samples were presented in Figure 3a, including surface coating, bonding layer and substrate. Various layers were separately shown in Figure 3b–d. As shown in Figure 3b–d, the thickness of the $Cr_3C_2$–NiCr coating was approximately 205 μm, and the porosity of the coating was 0.8%. In addition, all the samples had similar surface-coating features and entirely different bonding-layer features. What is more, a large amount of grey and dark lines was observed, which indicated a lamellar structure of the coating. This was ascribed to the particles to be a semi-melting state before arriving at the bonding layer and some semi-melting particles at the fringe of flame flow to be oxidation. Subsequently, the latter film covered on the deposited film, the oxide was trapped in the interlamination region of the surface coating layer. According to SEM images of the surface coating (Figure 4), a few porosities were detected. With respect to the porosity, it could be regarded as a stress concentration region under fatigue conditions. A large number of unmelted particles (red arrow) were evenly distributed in the bonding phase (blue arrow). According to the EDS results, the unmelted particle and the bonding phase were, respectively, proven to be $Cr_3C_2$ and NiCr. This particular microstructure can be ascribed to the fact that the input energy could not completely make the refractory $Cr_3C_2$ melting, whereas the energy was sufficient to melt down NiCr. Additionally, the carbide retention index usually ranged from 0.84 to 0.98 for HVOF technology [26], which implied that $Cr_3C_2$ has a strong metallurgical bond with NiCr. Thus, the tendency for stress concentration around the hard particles ($Cr_3C_2$) was subsequently alleviated.

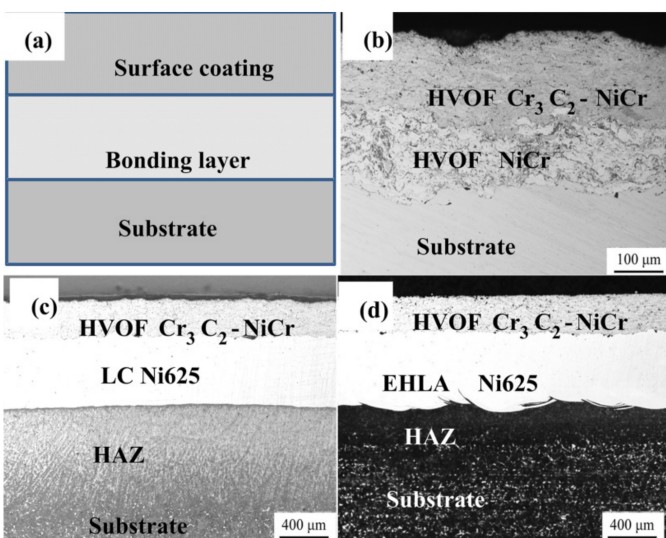

**Figure 3.** (**a**) The sketch of the complex coating; (**b**–**d**) presents the cross section of $Cr_3C_2$–NiCr + NiCr, $Cr_3C_2$–NiCr + LC Ni625 and $Cr_3C_2$–NiCr + EHLA Ni625.

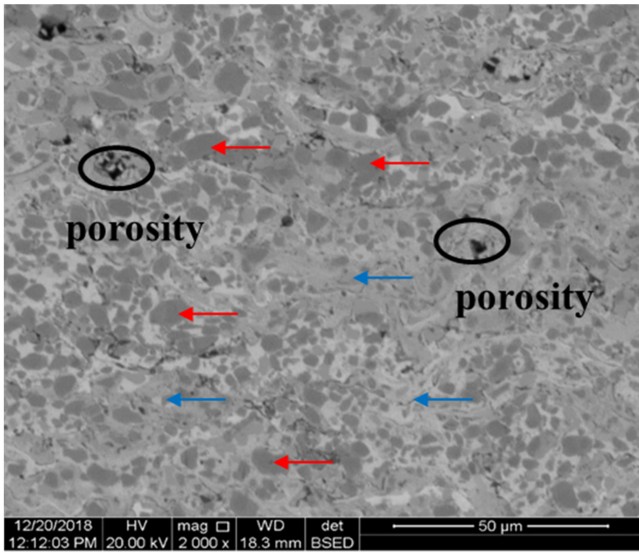

**Figure 4.** The microstructure of HVOF $Cr_3C_2$–NiCr surface coating.

The details of bonding layers were demonstrated in Figure 5. As shown in Figure 5a, the bonding layer of HVOF NiCr displayed similar feature to that of surface coating, whereas there is no visible porosity. The porosity of the NiCr coating was 0.2%. The occurrence of dark lines in the image had the same mechanism in comparison with that in the surface coating. The deposited bonding layer was around 100 μm. For the laser cladding layer, the microstructure of both layers fabricated by LC and EHLA had the composition of the γ-Ni and Laves phase [25], and the thicknesses of bonding layers were 412 μm and 399 μm under the given parameters, respectively. As shown in Figure 5b,c, the ELHA layer had a much finer crystal size than that of the LC layer. This observed difference indicated that the phase in HAZ of the LC sample had a slower solidification rate leading to enough driving force to crystalize comparing with that of the ELHA sample. This was because the scanning velocity of later was 7.5 times faster than that of the former. This phenomenon was strongly in accordance with previous studies [29]. Additionally, it also confirmed that the ELHA layer had more dislocations and stacking faults in comparison with the conventionally LC layer as the result of an ultrahigh cooling rate under high scanning velocity.

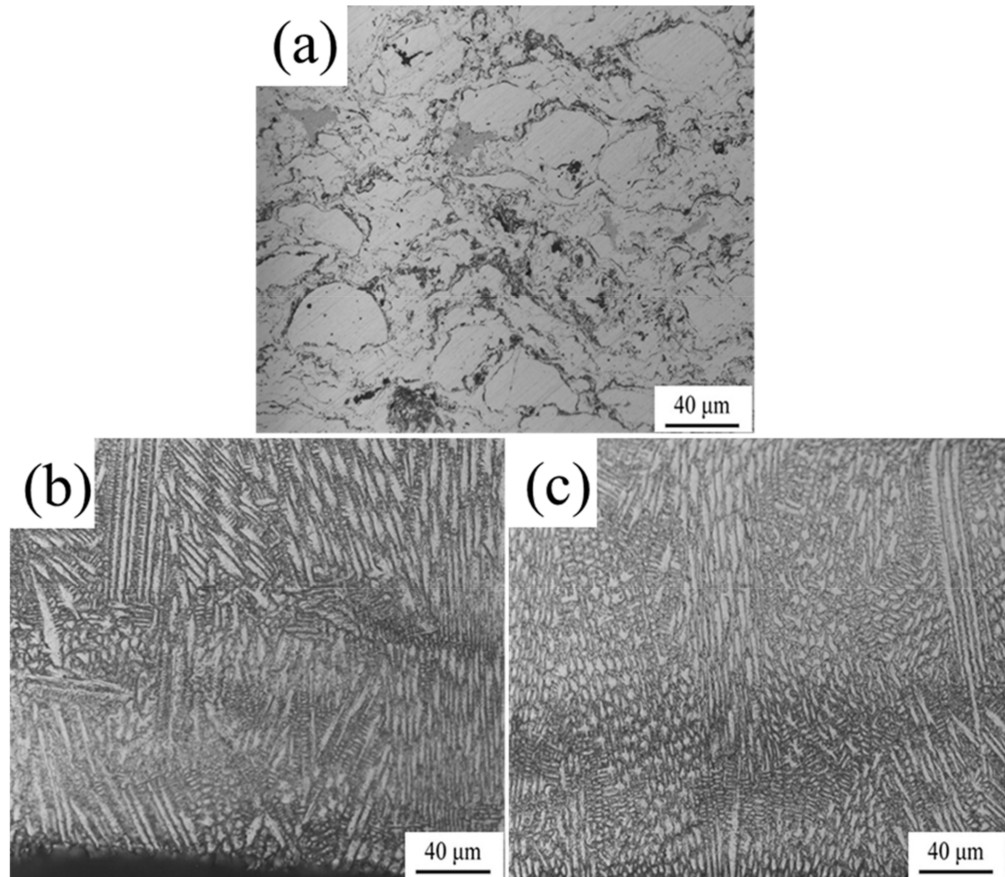

**Figure 5.** The microstructure of bonding layer: (**a**) HVOF NiCr; (**b**) LC Ni625 and (**c**) EHLA Ni625.

The details of the interfaces between bonding layers and substrates were shown in Figure 6. Figure 6a indicated the original microstructure which was composed of lamellate pearlite and latticed ferrite. As displayed in Figure 6b, an apparent boundary was discovered between the bonding layer and the substrate, which implied a mechanical bonding type. Additionally, no distinct change on the microstructure of the substrate was detected comparing with the original microstructure of matrix. Thereby, there was no so-called heat-affected zone (HAZ) for the HVOF technique. For the HAZ (Figure 6c,d), the microstructures of the LC and EHLA samples were significantly different in comparison with the original microstructures. The depth of the LC sample was much larger than that of the EHLA sample, whose depth magnitudes are 218 μm and 774 μm, respectively. Furthermore, in the high magnification images, the phase of LC sample's HAZ (Figure 6e) was apparent martensite, whereas that of the EHLA sample's HAZ (Figure 6f) was composed of a small quantity of martensite and a large amount of sorbite. In terms of the difference, the explanations were correspondingly given as follows: as it is known, the process of LC was to feed the molten metal powder into the molten base pool. As a rule of thumb, approximately 80% laser energy is absorbed by the substrate and the remaining energy was used to melt the metal powder for LC treatment. On the contrary, the metal powder was melted above the substrate surface and subsequently deposited in the EHLA treatment. Therefore, around 80% of the laser energy was devoted to melt the metal powder and the remaining laser energy was absorbed by the matrix. The temperature at HAZ of the LC sample rapidly attained to Ac3 as the high heat absorption and subsequently transformed to martensite. As a comparison, the absorbed heat by the ELHA sample HAZ was not sufficient for the microstructure to be fully transformed into austenite, but sufficient to convert to sorbite in large quantities and a small amount of martensite. In addition, the depth of the HAZ was proportional to the absorbed heat, which resulted in the distinct HAZ depths of the LC and EHLA samples.

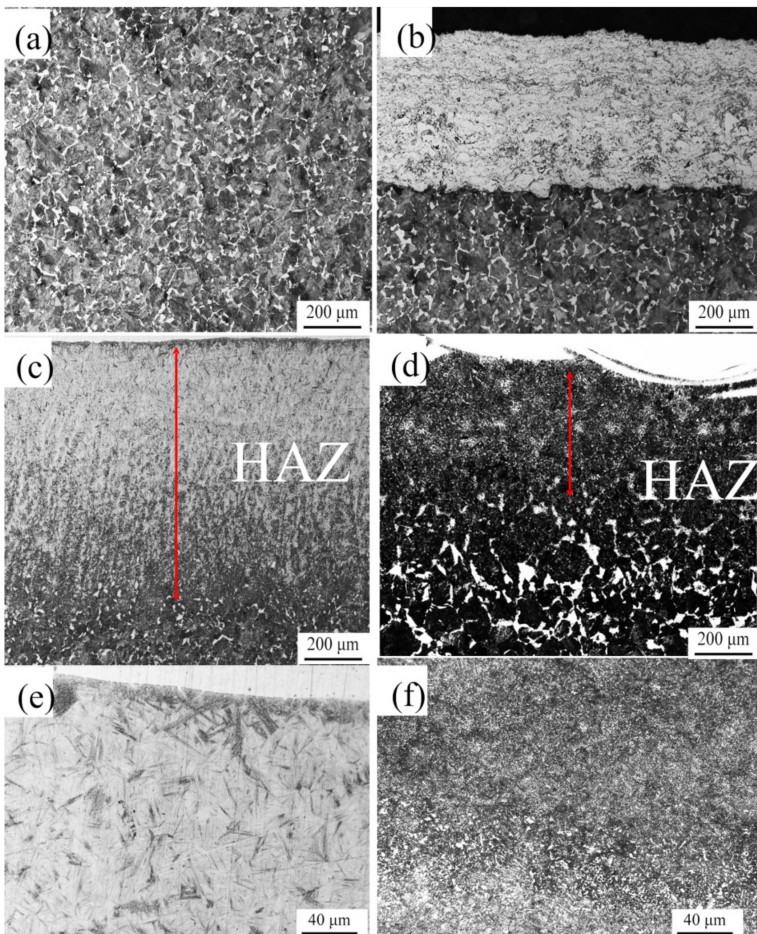

**Figure 6.** (**a**) the microstructure of substrate; (**b**–**d**) the cross section of No. 1–3; (**e**) and (**f**) present the HAZ details of the LC Ni625 and EHLA Ni625 layer.

Figure 7 plotted the hardness of the complex layer as a function of the distance from the top of surface coating. It was seen that the hardness of $Cr_3C_2$–NiCr (HVOF surface coating) was as high as 900 $HV_{0.3}$, owing to the uniform distribution of rigid $Cr_3C_2$ particles. For the bonding layer, the fabricated NiCr layer decreased to 408 $HV_{0.3}$ by the absence of rigid particles. Due to the low hardness of this layer, it had reasons to believe that it could not only provide enough toughness, but also eliminated potential stress concentration sources under fatigue condition. For the Ni625 layer fabricated by LC and EHLA, the magnitudes of hardness were 298 $HV_{0.3}$ and 345 $HV_{0.3}$, respectively. This was because the refinement of the EHLA layer microstructure resulted in a higher magnitude of hardness according to the Hall–Petch relationship [29]. On the other hand, the increase in dislocation density increased surface hardness as well. Regarding to the hardness of HAZ, it was notable that the plot of the LC sample rapidly arose to an average of 600 $HV_{0.3}$ (at HAZ), whereas the plot of the EHLA sample maintained a continuous downward trend. This difference was considered to be the result of the different HAZ microstructures, as previously mentioned.

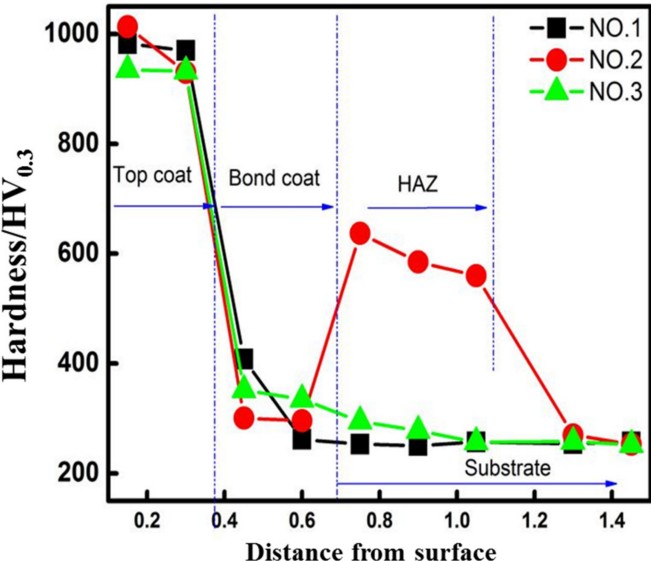

**Figure 7.** The hardness of samples as a function of the distance from the surface.

### 3.2. Dynamic Bend Tests

In this section, it was necessary to define interface 1 and interface 2 representing the boundary of surface coating and bonding layer, and the boundary of the bonding layer and the substrate, respectively.

Figure 8 demonstrated how the penetration testing surface profile of the sample suffered the BFT. Figure 8a indicated the sample with NiCr bonding layer after 500 cycles, where a large amount of intensive and successive macro cracks were presented. No apparent crack was found on the surfaces of samples No. 2 and No. 3. With further testing, the fatigue cracks emerged on the sample No. 2's surface at 1000 cycles (Figure 8b). After 1500 cycles, sample No. 3's surface manifested visible cracks, as presented in Figure 8c. As a comparison, the density (Figure 8a–c) and width (Figure 8d–f) of the cracks on the samples were sorted as: No. 1 > No. 2 > No. 3. Although sample No. 1 underwent the least amount of testing time, the surface damage was much greater than that of No. 2 and No. 3. It can be concluded that No. 1 presented the worst fatigue resistance, followed by No. 2.

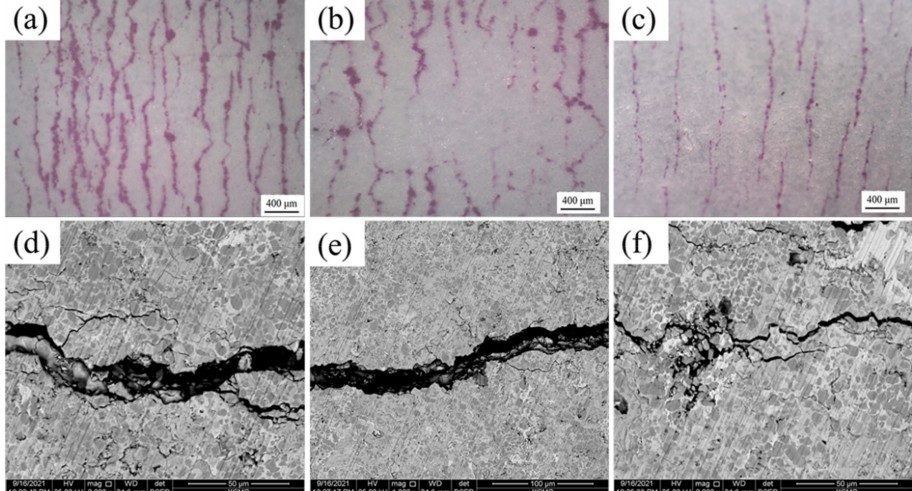

**Figure 8.** (**a**–**c**) the penetrating tested surface of No. 1–3 after BFT and (**d**–**f**) the high magnification image of fatigue crack.

In order to investigate the details of the failure, three cross sections were displayed in Figure 9. With respect to Figure 9a, the delamination was correspondingly formed after

the BFT because of the fragile surface. Due to the porous and laminar complex layer, the perforative cracks were prone to be initialized under fatigue condition, which directly exposes the substrate in the air. In addition, cracks were detected at the interface 2 as the poor adhesion with substrate. For sample No. 2 (Figure 9b), not only were cracks formed in the surface coating, but also interior cracks were discovered in the bonding layer. It should be noted that the mentioned cracks were not connected. This was because the cracks initialize from different resources: surface coating and the bonding layer. Additionally, referring to the red circle marked in the image, when the cracks from surface coating arrive at the bonding layer, the mechanical interface deflected it along interface 1. Although cracks were found in the bonding layer and the surface coating, the bonding layer was believed to act as a hinder layer because no perforative crack appears. Therefore, the behavior of fatigue cracks for the complex coating could be concluded as follows. The fatigue cracks occurred in both layers after bending test. As the cycle progresses, the cracks propagated toward to the interface 1. When one crack from one layer arrived at the interface 1, it propagated along the interface 1. When the other crack initialized from the other layer arrives at the interface 1, both cracks connected and propagated toward to the interface 2. Thus, the perforative crack would form at some point in the future, resulting in an exposed substrate. On the other hand, if adjacent cracks in surface coating connected at interface 1 before the crack in bonding layer arriving surface, the delamination would be preferentially incurred.

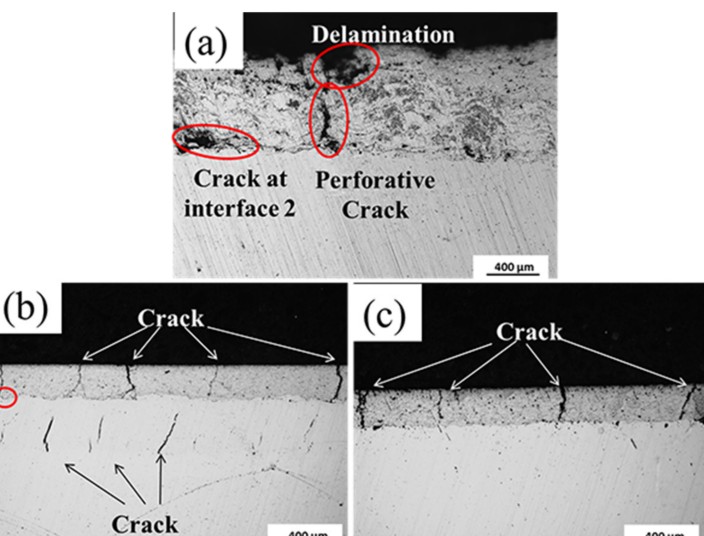

**Figure 9.** The cross section of the complex coating after BFT: (**a**) No. 1; (**b**) No. 2 and (**c**) No. 3.

Regarding sample No. 3 (Figure 9c), cracks were only detected in the surface coating and no apparent crack is found at interface 1. This could well state that the EHLA Ni625 bonding layer can evidently decrease the driving force of fatigue cracks in surface coating due to high toughness. Furthermore, as Shen et al. [30] pointed out, the refined crystal of the EHLA Ni625 layer had high density of grain boundary and dislocation, which effectively delays formation of the fatigue cracks. Hence, no crack formed at the interior of the bonding layer. There was no clear difference between sample No. 2 and sample No. 3. The preferred failure mode could be delamination for sample No. 3 and perforative cracks for sample No. 2 as the cycle progressed.

### 3.3. Electrochemical Behavior Analysis

The electrochemical behavior of the coatings was investigated by measuring potentiostat polarization curves and EIS. Potentiodynamic polarization curves of samples No. 1, No. 2 and No. 3 coatings during 500 h of corrosion at −350 mV were tested, as shown in Figure 10. There were fluctuations in testing process of current density. The results

indicated that final current density of samples No. 1–No. 3 ranged from $-1.4\ \mu A/cm^2$ to $-1.2\ \mu A/cm^2$. A negative net current showed a cathodic behavior of the coatings and no porosity and micro-cracks penetrating the coating down to the substrate. There was little difference in these three curves, meaning that the corrosion resistance of different coatings under the action of pure corrosion was close. The negative net current indicated the cathodic behaviors of the coatings.

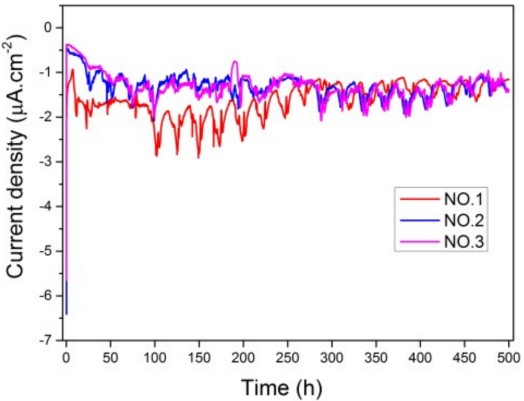

**Figure 10.** The curves of potentiostatic polarization curve of No. 1–No. 3 without BFT.

Figure 11 showed the Bode plots of the samples before and after the dynamic bend test. From the data of Figure 11a, both total impedances ($|Z|$) of samples No. 2 and No. 3 without the dynamic bend test were much higher than that of sample No. 1 at 0.01 Hz. It was proven that the interfacial charge transfer of samples No. 2 and No. 3 was more difficult. As observed in Figure 11b, in the intermediate frequency range of samples No. 2 and No. 3, log$|Z|$ increased linearly as logf decreases. Moreover, it can be seen that the slope of the line is close to negative 1, which implied that no corrosion occurs. This can be attributed to barrier properties of the bonding layer. In the phase angle-log curve, the impedance diagrams of samples No. 2 and No. 3 in Bode phase angle presented a broad peak, indicating excellent density and barrier characteristics.

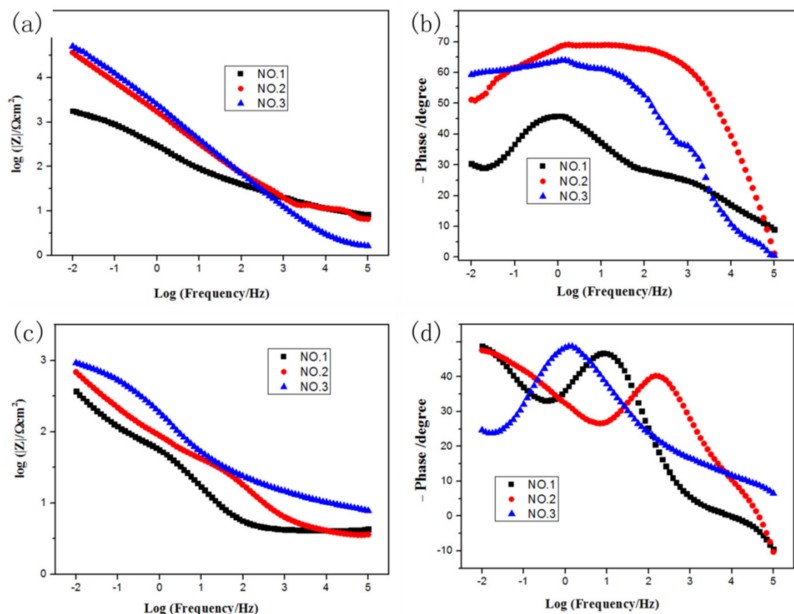

**Figure 11.** Bode plots recorded of the sample without BFT: (**a**) lg $|Z|$-lgf; (**b**) phase-lgf; Bode plots recorded of the sample after BFT: (**c**) lg$|Z|$-lgf (**d**) phase-lgf.

As shown in Figure 11c, the total impedance (|Z|) at 0.01 Hz of the samples No. 3 and No. 2 reduced from ~$10^5$ to ~$10^3$ after BFT. The total impedance (|Z|) of at 0.01 Hz the samples No. 1 reduced from ~$10^{3.5}$ to ~$10^{2.5}$ for the sample No. 1 after BFT. It was proven that the interfacial charge of the three samples was prone to transfer after BFT. Furthermore, the total impedance (|Z|) at 0.01 Hz from relatively high to low sequence was the sorted as No. 3 > No. 2 > No. 1. The results illustrated the interfacial charge transfer of sample No. 3 was most difficult after BFT. As displayed in Figure 11d, in the phase angle-logf curve, the impedance diagrams of No. 1 and No. 2 samples in Bode phase angle show two obvious peaks. There were predominately two time constants, where one is the inductance arc and the other one was the contain arc. It implied that the corrosion solution reached the interface 2 through the perforative micro-crack formed during the dynamic bend test of samples No. 1 and No. 2. However, this phenomenon did not occur in sample No. 3 due to the excellent corrosive resistance of the EHLA layer.

### 3.4. Salt Spray Corrosion Tests

Owing to the excellent corrosive resistance of the surface coating ($Cr_3C_2$–NiCr HVOF layer), a small quantity of pitting was found until the NSS test time reached 3000 h for all samples without BFT. The corrosive morphology was shown in Figure 12. In order to further study the corrosion behavior, the EDS was executed on pitting, and the results were shown in Figure 12 and Table 5. As listed, no Fe element was detected and some O element was found, which proves that the corrosion occurred at the Ni–Cr phase.

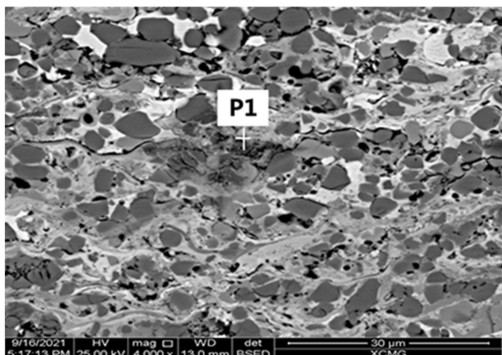

**Figure 12.** The corrosive surface morphology (3000 h) of all the samples without BFT.

**Table 5.** The composition of chemical elemental in weight %.

| Element | C | O | Cr | Mn | Ni |
|---------|------|------|-------|------|-------|
| Content | 1.63 | 7.17 | 62.07 | 6.53 | 22.61 |

As the $Cr_3C_2$–NiCr coating contained amount of Cr element, the pitting corrosion resistance was apparently superior, according to the PREN factor. The dominant corrosion mechanism of $Cr_3C_2$–NiCr coating exposed to artificial seawater can be divided into three stages. (i) $Cr_2O_3$ and NiO passive film can be formed on the surface of the $Cr_3C_2$–NiCr coatings. (ii) $Cl^-$ penetrated through the thin passive layer at the passive film/surface coating interface. (iii) $Cl^-$ penetrated through thin passive layer at the surface coating/bonding layer interface. According to the results, it was supposed that the surface coating and bonding layer acted as a barrier delaying the attack of the substrate; the surface coating thickness of 0.2 mm and low porosity were the important factors.

For sample No. 1 suffering the BFT, corrosive morphologies (Figure 13a) were demonstrated in the form of broken surface and formation of through-wall micro-cracks. To identify the corrosion products, EDS was operated on the surface. As the results shown (Figure 13b–d), there was almost 100% Fe element covering the severely corroded area (white rectangle in Figure 13a). This was caused by the fact that the substrate was directly

exposed in the corrosive environment after BFT. The ferric oxide overflew though the perforative crack and distributed nearby the cracks. The corrosion behavior in other area was similar to the sample without BFT.

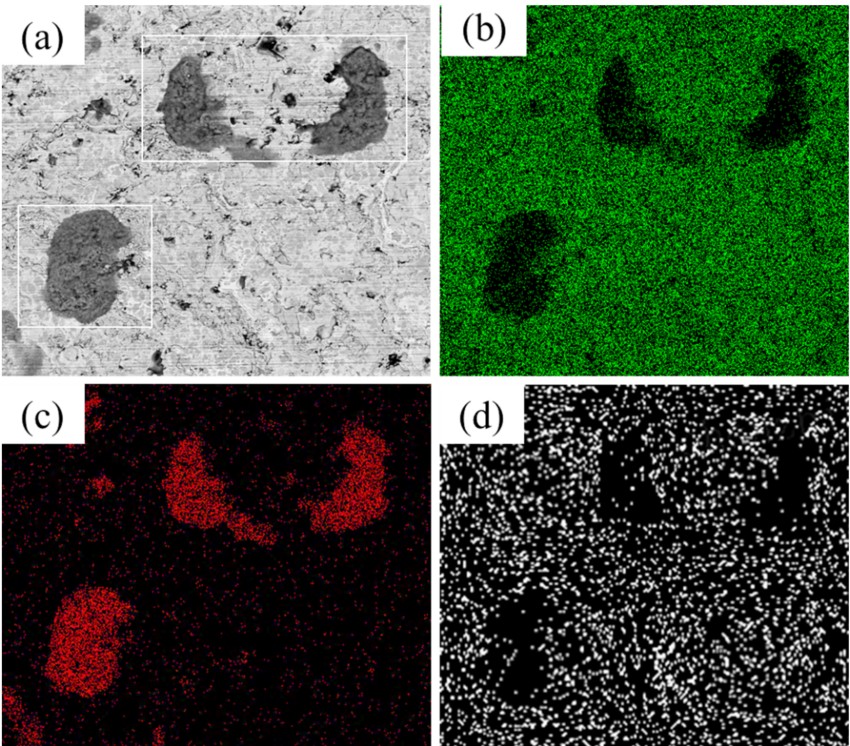

**Figure 13.** (**a**) The corrosive surface of No. 1 suffered BFT and (**b**–**d**) Mapping EDS of Cr, Fe and Ni elements.

After BFT, the corrosive morphologies of samples No. 2 and No. 3 were similar to the surface profiles of the samples without BFT. It seemed that the surface failure was not accelerated under a combination of corrosion and bending fatigue conditions for the LC and EHLA samples. This was because a large number of cracks appeared on the BFT samples surface, but these cracks did not cross the surface coating and bonding layer. Thus, the laser cladding layer played a significant role in the obstruction of corrosive elements. The main elements involved in corrosion were Ni and Cr, which were the same as those samples without BFT. However, it can be seen in Figure 14 that a large number of tiny cracks formed on the surface in comparison with the sample not undergoing BFT. With respect to the surface profile, the magnitude and width of the micro-cracks presented in Figure 14b were apparently less than those of displayed in Figure 14a. In terms of the formation of these micro-cracks, it was proposed that the corrosion mechanism was stress corrosion, which is attributed to the BFT [31]. The phenomenon could well explain that the impedance modulus decreased by 1–2 orders of magnitude for the BFT samples, comparing with the samples without BFT. As BFT and NSS were not carried out at the same time, the process of surface failure under the combination of corrosion and bending fatigue condition could be dependably speculated. BFT initializes the cracks and stress concentrations, which promoted corrosion behavior. Once Ni and Cr oxides appear, the passivation coating of NiO and $Cr_2O_3$ would be easily broken under BFT (Figure 15) and etch pits would form as a result. Thus, the stress concentration process was correspondingly accelerated under fatigue conditions due to these etch pits. As the corrosion and fatigue progress, the surface failure was developed to the interior of the layer. As a conclusion, the corrosion and fatigue behavior would be significantly promoted by each other, leading to surface failure.

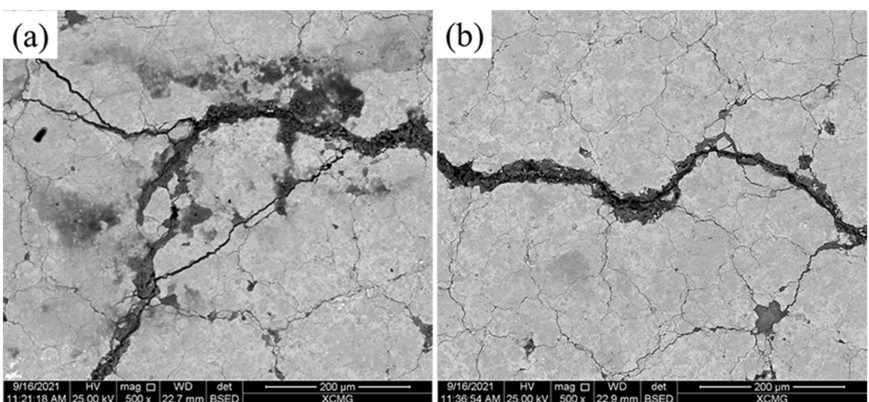

**Figure 14.** The corrosive surface of (**a**) No. 2 and (**b**) No. 3 suffered BFT.

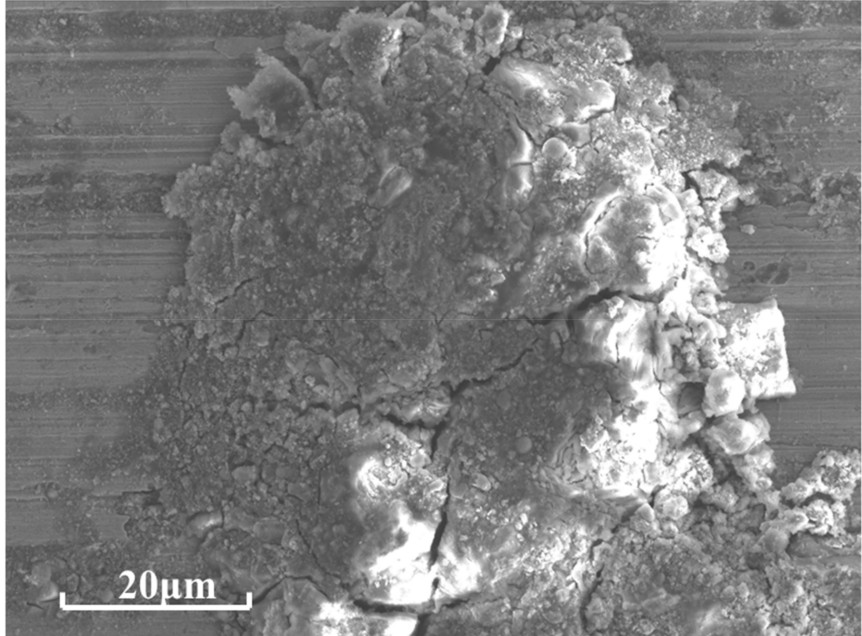

**Figure 15.** Broken passivation coating of NiO and $Cr_2O_3$ corrosion products.

In the present study, the failure modes of samples No. 2 and No. 3 were similar under the combination of fatigue and corrosion conditions. Due to the differences between EHLA Ni625 and LC Ni625 bonding layers, the failure behavior of both samples should be remarkably different due to the different distribution of interior cracks. As proven above, the corrosion behavior was developed depending on the macro-cracks, which would lead to different failure modes for samples No. 2 and No. 3. Nevertheless, due to time restrictions, the exact failure mechanism of the samples with EHLA Ni625 and LC Ni625 bonding layers were not fully distinguished from each other, and it will be examined further in our future research.

## 4. Conclusions

1. The cycles of perforative cracks for the sample with the EHLA Ni625 bonding layer were almost three timeslarger than the sample with the HVOF NiCr layer.
2. The magnitude of EIS reduces from ~$10^5$ to ~$10^3$ for the sample before and after BFT, which is ascribed to the occurrence of fatigue cracks.
3. The main mechanism of service life promotion under the fatigue and corrosion coupling environment is an improved bonding layer, including refinement grain, good metallurgical bonding with substrate and less interior defects (porosity, crack and unmelted particle).

**Author Contributions:** L.Z. and X.Y. (Xiaoming Yuan) clearly defined the research ideas and research methods. X.Y. (Xiao Yun), J.W. and G.Z. are responsible for completing the experimental part of the paper, B.H. is responsible for writing the paper, and Z.C. is responsible for revising the paper and experimental methods. All authors have read and agreed to the published version of the manuscript.

**Funding:** This research was funded by the National Key Technologies Research and Development Program, the number is 2019YFB2005301 and the Natural Science Foundation of Hebei Province the number is No. E2020203090.

**Institutional Review Board Statement:** Not applicable.

**Informed Consent Statement:** Not applicable.

**Data Availability Statement:** The datasets generated during and/or analyzed during the current study are available from the corresponding author on reasonable request.

**Conflicts of Interest:** The authors declare no conflict of interest.

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
