# Peer review of "Comparative Study of HVOF Cr3C2–NiCr Coating with Different Bonding Layer on the Interactive Behavior of Fatigue and Corrosion"

_coatings, doi:10.3390/coatings12030307_

Round 1

Reviewer 1 Report

This manuscript studies the fatigue and corrosion performance of an HVOF Cr3C2-NiCr coating under different bonding layers. The following points should be considered, and I am interested to reconsider the manuscript after a major revision:

1- About the Bode plot results, more explanation is needed. How you can relate the results with the mechanical properties? Characteristics of the graphs should be discussed.

2- The introduction section is not enough attractive for the readers. The authors should explain about the important points related to this study in the introduction. There are some studies about coatings showing the effect of grain refinement on the corrosion properties. This should be explained in the introductory part, and these references should be cited:

https://doi.org/10.1016/j.elecom.2021.107169 https://doi.org/10.1016/j.surfcoat.2014.11.044 

3- The manuscript was written in rush, and there are some typos and errors. The language of the manuscript should be revised by an English native speaker who is an expert in the field.

4- What is this?! 'Authors should discuss the results and how they can be interpreted from the perspective of previous studies and of the working hypotheses. The findings and their implications should be discussed in the broadest context possible. Future research directions may also be highlighted.'

5- In line 355, it was explained that stress corrosion cracking is the mechanism of the corrosion attributed to BFT. There are several indications in fractured surface, and stress distribution to prove that a phenomenon is SCC. This important part should be explained by literature in a logical way.

6- The authors also mentioned passivation, however, depending on the samples (1, 2, 3), the passivation procedure should be explained. Also, using the PREN factor can be insightful. 

Reviewer 2 Report

  1. The whole abstract needs to be rewritten. The significance and purpose of this research should be clearly presented in the abstract. The abstract must be presented in a clear way in problematic, objective, idea, description of idea, highlighting the methods, results, quantitative comparison of results with significant findings, conclusions.
  2. Revise the introduction section. The state-of-the-art comparisons for the proposed work are missing in this paper. Then do a critical analysis of previous research. State explicitly the shortcomings of previous research. What is positive in previous research and what is negative. Based on that, you explicitly define the goal of the research and the scientific hypothesis.
  3. Highlight the novelty of your methodology.
  4. What is the adhesion strength of coating?
  5. What is the degradation rate of coatings in terms of weight?
  6. The biggest shortcoming of the research is that there is no analysis of errors, analysis of the sensitivity of results and analysis of uncertainty of results.
  7. The Conclusion section should be rewritten. Highlight your scientific contribution. Highlight the benefits of your research. Define shortcomings and future research.
  8. What about the wear resistance of coatings? Authors must discuss the same.

Reviewer 3 Report

The submitted manuscript requires significant revision:

1) In the section "Introduction" in works [3–5], composite materials are described, but of a different composition than yours. Are there published data of the Cr3C2–NiCr coatings? What is remarkable about this composite material?

2) It may be necessary to supplement the “Introduction” section, as 15 literature references in the submitted manuscript are completely insufficient.

3) Figs. 3c, 3d, 6d, 13 are of bad quality, please fix them.

4) Has the porosity of the presented deposited intermediate layers and the coatings been evaluated? For the HVOF process, the porosity criterion is critical.

5) Fig. 7. Why is there no increase in hardness in the HAZ zone for sample No. 3? Explain.

6) There are typos.

Fig. 1 - repeated twice Fig. 1d.

Line 104 - HOVF process. Please fix.

Reviewer 4 Report

1.The subject of the article is interesting. In my opinion, the title corresponds to the content. The introduction should be extended to include more references.
2. The chemical composition of the substrate is better to be presented in the table.
3. Please check and improve text (punctuation, no spaces, double spaces, typos etc.)
4. Why were 2 types of laser used. Was it possible to change the spot size with one device?
5. According to which standard the microhardness was measured. Most standards specify microhardness up to 200 g. 300 g was used in the paper, which is a value corresponding to the hardness tests, not the micro-hardness tests.
6. A note on aesthetics. Everyone has a different aesthetics, so it's not a meaningful comment. However, the inscriptions on the microstructures in Fig. 3 are very large. A smaller one would suffice.
7. In Fig. 4 "hole" is written. In my opinion it should be porosity
Please enlarge Fig. 7.
8. Fig. 13 should be called Mapping EDS of ...
9. Please put a space between Fig and the number
10. Number 6 Patents? What is it about?
Major revision 

Reviewer 5 Report

This paper studied the anti-fatigue and anti-corrosive performance of HVOF Cr3C2-NiCr coatings with different bonding layers. The work is comprehensive while part of the manuscript is not of good quality. Following comments are provided to be considered for improving it.

1) All the figures are in low resolution, please provide better figures.

2)  Most of the microstructure images are with low contrast, which results in bad quality.  Such as Figure 3, Figure 4, Figure 6,...

3) Please provide the detailed distribution of the powder size.

4) In L127, why do you want to achieve the 90% of theoretical yield strength of the substrate. 

5) In L 131, the authors stated that three repetitive tests were conducted, while no data derivation has been provided.

6) In L 134, how can you determine "if a crack was detected"?

7) For Figure 6(f), nothing can be obtained from this image.

8) For Figure 7, hardness should be tested for at least three data points for each location since it may have a large deviation due to the underlying microstructure. 

9) In L 253, it is too arbitrary to conclude that Sample 1 has the worst fatigue resistance.

10) The information in Figure 10 is unclear. 

11) There is no scale bar in Figure 13.

Round 2

Reviewer 1 Report

The revised version is far better than the previous version. However, it still needs an English language revision and checking the typos. 

Author Response

We are very sorry that the English is not good enough and we tried our best to improve the whole paper. Moreover, we have resorted to help from native English speakers for improvement of the language once again and the revised section is highlighted in the paper. Therefore, we sincerely hope the revised version could suffice the requirement for consideration of publication.

Reviewer 3 Report

It seems to me that 21 references are not enough for a sufficient literature review.

Author Response

Thanks for your kindly comment. According to the suggestion, we have added the relative references in present paper. The total reference attains 32.

Reviewer 5 Report

The response to my comment is fine

Author Response

Thanks for your kindly comment.